# High-throughput, single-copy sequencing reveals SARS-CoV-2 spike variants coincident with mounting humoral immunity during acute COVID-19

Sung Hee Ko[1,☯], Elham Bayat Mokhtari[1,☯], Prakriti Mudvari[1,☯], Sydney Stein[2,3], Christopher D. Stringham[1], Danielle Wagner[1], Sabrina Ramelli[2], Marcos J. Ramos-Benitez[2,3], Jeffrey R. Strich[2], Richard T. Davey, Jr.[3], Tongqing Zhou[1], John Misasi[1], Peter D. Kwong[1], Daniel S. Chertow[2,3], Nancy J. Sullivan[1], Eli A. Boritz[1]*

1 Vaccine Research Center, National Institute of Allergy and Infectious Diseases, National Institutes of Health, Bethesda, Maryland, United States of America, 2 Emerging Pathogens Section, Critical Care Medicine Department, National Institutes of Health Clinical Center, Bethesda, Maryland, United States of America, 3 Laboratory of Immunoregulation, Division of Intramural Research, National Institute of Allergy and Infectious Diseases, National Institutes of Health, Bethesda, Maryland, United States of America

☯ These authors contributed equally to this work.
* boritze@niaid.nih.gov

**Data Availability Statement:** Raw PacBio CCS sequence data associated with this study have been deposited in the NCBI SRA database with the

## Abstract

Tracking evolution of the severe acute respiratory syndrome coronavirus 2 (SARS-CoV-2) within infected individuals will help elucidate coronavirus disease 2019 (COVID-19) pathogenesis and inform use of antiviral interventions. In this study, we developed an approach for sequencing the region encoding the SARS-CoV-2 virion surface proteins from large numbers of individual virus RNA genomes per sample. We applied this approach to the WA-1 reference clinical isolate of SARS-CoV-2 passaged *in vitro* and to upper respiratory samples from 7 study participants with COVID-19. SARS-CoV-2 genomes from cell culture were diverse, including 18 haplotypes with non-synonymous mutations clustered in the spike $NH_2$-terminal domain (NTD) and furin cleavage site regions. By contrast, cross-sectional analysis of samples from participants with COVID-19 showed fewer virus variants, without structural clustering of mutations. However, longitudinal analysis in one individual revealed 4 virus haplotypes bearing 3 independent mutations in a spike NTD epitope targeted by autologous antibodies. These mutations arose coincident with a 6.2-fold rise in serum binding to spike and a transient increase in virus burden. We conclude that SARS-CoV-2 exhibits a capacity for rapid genetic adaptation that becomes detectable *in vivo* with the onset of humoral immunity, with the potential to contribute to delayed virologic clearance in the acute setting.

## Author summary

Mutant sequences of severe acute respiratory syndrome coronavirus-2 (SARS-CoV-2) arising during any individual case of coronavirus disease 2019 (COVID-19) could theoretically enable the virus to evade immune responses or antiviral therapies that target the

BioProject accession number PRJNA680710. The
bioinformatic analysis pipeline developed in this
study is available at https://github.com/niaid/UMI-
pacbio-pipeline.

**Funding:** Funding was provided by the Intramural
Research Program of the U.S. National Institutes of
Health (project AI005157-01, E.A.B). M.J.R-B. is
supported by NIGMS Postdoctoral Research
Associate Training Program (1FI2GM137804-01).
The funders had no role in study design, data
collection and analysis, decision to publish, or
preparation of the manuscript.

**Competing interests:** The authors declare no
competing interests.

predominant infecting virus sequence. However, commonly used sequencing technolo-
gies are not optimally designed to detect variant virus sequences within each sample. To
address this issue, we developed novel technology for sequencing large numbers of indi-
vidual SARS-CoV-2 genomic RNA molecules across the region encoding the virus surface
proteins. This technology revealed extensive genetic diversity in cultured viruses from a
clinical isolate of SARS-CoV-2, but lower diversity in samples from 7 individuals with
COVID-19. Importantly, concurrent analysis of paired serum samples in selected individ-
uals revealed relatively low levels of antibody binding to the SARS-CoV-2 spike protein at
the time of initial sequencing. With increased serum binding to spike protein, we detected
multiple SARS-CoV-2 variants bearing independent mutations in a single epitope, as well
as a transient increase in virus burden. These findings suggest that SARS-CoV-2 replica-
tion creates sufficient virus genetic diversity to allow immune-mediated selection of vari-
ants within the time frame of acute COVID-19. Large-scale studies of SARS-CoV-2
variation and specific immune responses will help define the contributions of intra-indi-
vidual SARS-CoV-2 evolution to COVID-19 clinical outcomes and antiviral drug
susceptibility.

## Introduction

Although SARS-CoV-2 genetic diversification was initially slow as the virus spread around the
world [1], the extent and implications of intra-individual virus evolution during COVID-19
are still being explored. Close genetic relationships among single-person SARS-CoV-2 consen-
sus sequences do not rule out intra-individual evolution because virus burden and transmissi-
bility peak shortly after acquisition [2–4], before the development of adaptive immune
responses that could select transmissible virus variants. Furthermore, SARS-CoV-2 evolution
has been detected in people with compromised immunity, with shifts in virus consensus
sequences noted during prolonged shedding [5–9]. In early infection, however, analysis of
SARS-CoV-2 sequences has not routinely demonstrated directional genetic change. Sites in
the virus genome showing significant intra-individual variation have been found in cross-sec-
tional data [10–14], with one study linking the number of variant sites to disease severity at the
time of study [15]. Nonetheless, studies in clinically diverse cohorts have found that SARS-
CoV-2 consensus sequences [16] and minor variants [14] remain stable in most people over
time. These findings would suggest that immune responses against transmitted virus strains
should continue to target replicating viruses throughout the course of each individual's
infection.

   An important obstacle to understanding intra-individual evolution of SARS-CoV-2 is that
standard sequencing and analytical procedures yield a single consensus sequence for each sam-
ple, rather than multiple sequences representing virus quasispecies diversity. Standard proce-
dures typically either amplify virus RNA in fragments spanning the genome or produce meta-
transcriptome libraries of fragments from the entire sample [17], followed by short-read deep
sequencing, read alignment or assembly, and virus genome consensus determination. These
approaches readily cover nearly the entire 30-kilobase length of the SARS-CoV-2 genome for
samples from hundreds or thousands of people at a time, helping to define inter-individual
virus variation on a global scale [1]. However, combined amplification from multiple genomes
and the "shotgun" sequencing of long regions in small fragments can both disrupt genetic link-
age and prevent error correction at the level of individual haplotypes. Analysis of intra-individ-
ual variation within resulting data is thus largely limited to the detection of genome positions

at which variation occurs at levels exceeding the background arising from sample preparation and sequencing errors. As a result, standard methods could miss important patterns of intra-individual SARS-CoV-2 diversity and evolution due to insufficient discrimination of true signal from technical noise.

In this report we use a single-genome amplification and sequencing (SGS) approach to investigate the genetic diversity of SARS-CoV-2 in samples from people with COVID-19. Our approach is conceptually similar to conventional SGS procedures, which amplify single molecules at limiting dilution for Sanger sequencing [18, 19]. However, to obtain a broad view of the SARS-CoV-2 variant pool, we developed a high-throughput SGS (HT-SGS) strategy employing long-read deep sequencing of the surface protein gene region from large numbers of individual virus genomes. Our results demonstrate the emergence of SARS-CoV-2 genetic variants under host immune pressure during acute infection.

## Results

### Validation of HT-SGS for SARS-CoV-2 surface protein gene sequencing

We developed an HT-SGS approach for sequencing individual virus RNA genomes within each sample across the spike (S), ORF3, envelope (E), and membrane (M) protein genes. This approach employs unique molecular identifier (UMI) tags added to the virus genome complementary DNA (cDNA) during reverse transcription (RT; Figs 1A and S1), and incorporates several layers of error correction in a custom bioinformatic pipeline (Figs 1A and S2). These include (i) consensus formation from reads with matching UMIs to remove PCR errors and those sequencing errors not addressed by circular consensus sequence (CCS) correction [20], (ii) initial removal of UMI bins with outlying low read counts by inflection point filtering (S2B Fig), (iii) network-based filtering to exclude false UMI bins arising from PCR or sequencing errors in the UMI (see Materials and Methods), and (iv) stringent removal of UMI bins with low read counts by knee point filtering (S2C Fig). Errors occurring during the RT step are then addressed by (v) flagging unique and potentially spurious insertions/deletions (indels) and other rare mutations by variant calling, for reversion to the sample consensus, and (vi) exclusion of sequence haplotypes occurring in only 1 UMI bin (i.e., unique SGS).

To validate our method, we applied it to clonal RNA transcripts representing the USA/WA-1 sequence (wt) or a double-mutant (2M) sequence that included two scrambled 20-base sections at the ends of the target region (Fig 1B). Using UMI bin consensus sequences obtained after knee point filtering, we calculated error rates of 0.00024/base for the wt target and 0.00025/base for the 2M target. No inter-template recombinants were detected. Putative errors included both single-nucleotide substitutions and short indels, and likely represented a combination of RT, PCR, and sequencing errors as well as *in vitro* transcription errors and plasmid mutations. After a completed analysis including variant calling, rare mutation reversion, and exclusion of unique SGS, we found that all remaining sequences exactly matched their corresponding references, with quantitative recovery of the two targets from a dilution series (Table 1). These results support the high accuracy of our data generation and analytical approach.

### HT-SGS analysis of a cultured clinical isolate of SARS-CoV-2

To begin evaluating intra-sample diversity of SARS-CoV-2, we applied our HT-SGS process to a 4[th]-passage Vero cell culture of the WA-1 reference clinical isolate. As shown in Fig 2A, the consensus of all HT-SGS sequences from this sample exactly matched the WA-1 reference sequence, consistent with the high accuracy of the method. However, data analysis at the single-genome level revealed 18 unique SARS-CoV-2 haplotypes detected in between 3 and 174

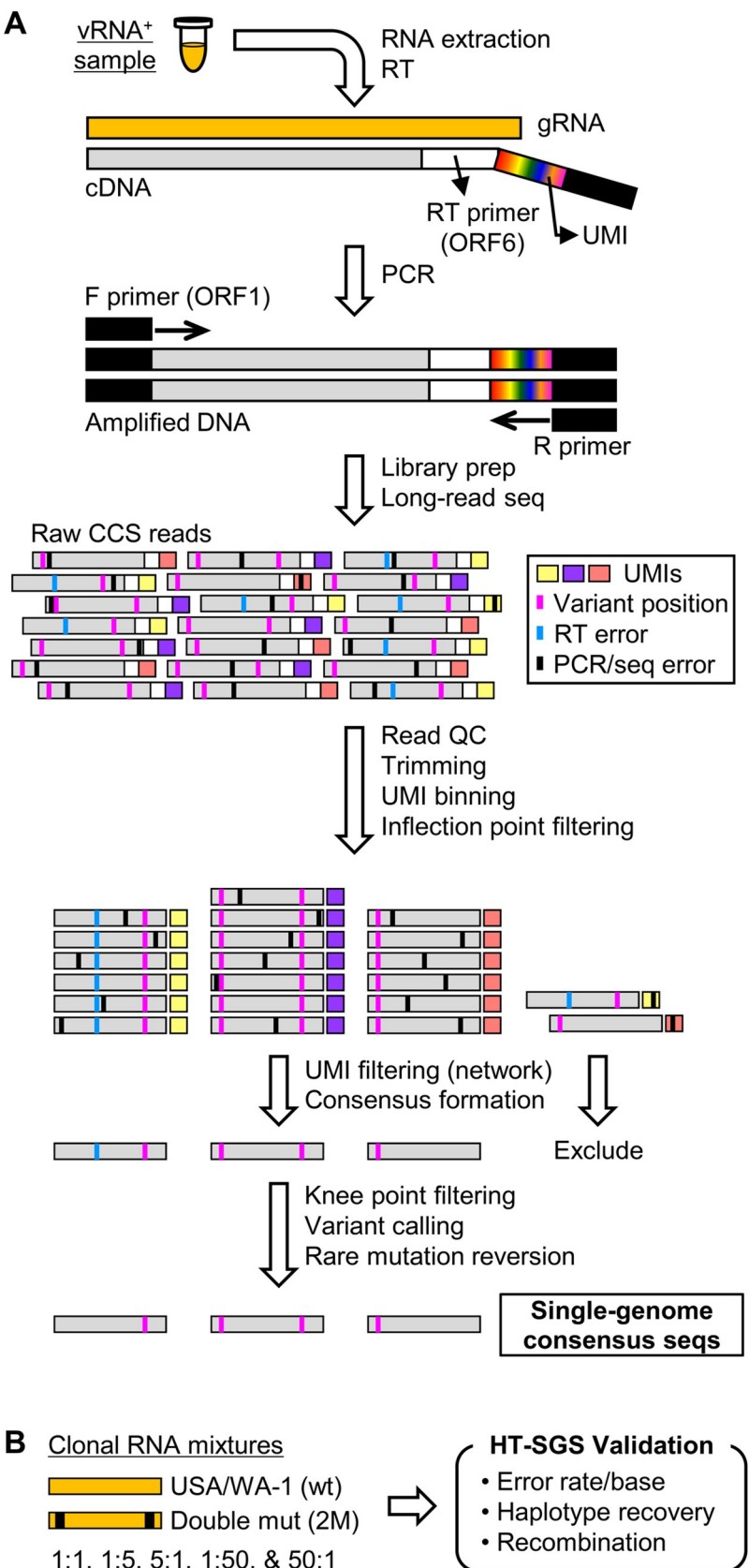

**Fig 1. Overview of HT-SGS data generation and analysis.** (A) SARS-CoV-2 genomic RNA (gRNA) is reverse-transcribed to include an 8-nucleotide unique molecular identifier (UMI; multicolored bar), followed by PCR amplification and Pacific Biosciences single-molecule, real-time (SMRT) sequencing of the 6.1-kilobase region encompassing spike (S), ORF3, envelope (E), and membrane (M) protein genes. After quality control and trimming, sequence reads are compiled into bins that share a UMI sequence, and bins with low read counts are removed according to the inflection point of the read count distribution (see S2B Fig). Presumptive false bins arising from errors in the UMI are then identified and removed by the network adjacency method, followed by further removal of bins with the lowest read counts using a more conservative knee point cutoff (see S2C Fig). Variant calling is then used to identify presumptive erroneous mutations based on rarity and pattern (ex., single-base insertions adjacent to homopolymers), and these are reverted to the sample consensus. Finally, SGS that correspond to haplotypes occurring only once in each sample are excluded (not pictured). (B) To validate data generation and analysis procedures, clonal RNAs transcribed *in vitro* from USA/WA-1 and double mutant sequences were mixed at varying ratios and subjected to HT-SGS. Results are described in Results and Table 1.

individual virus genomes per haplotype, with each single-genome consensus supported by >500 sequence reads (Fig 2A and 2B). More than half (57.6%) of all SGS differed from the reference consensus sequence at one or more nucleotide positions (Fig 2A). All 17 mutations detected in variant virus genomes were non-synonymous, suggesting selective pressure on the virus. Structurally, mutations were clustered almost exclusively in the spike NTD and furin cleavage site regions. The NTD mutations included 9 distinct single-nucleotide variants (SNVs) and 2 distinct insertions that added positively-charged or removed negatively-charged amino acid residues at the NTD outer surface (Fig 2A and 2C), consistent with observed selection patterns in other virus envelope proteins during cell culture adaptation [21, 22]. Mutations in the area of the furin cleavage site included 3 SNVs and one deletion of 12 amino acids (Fig 2A and 2C), and were consistent with mutations observed in this region after *in vitro* passage in other studies [23]. The remaining 2 mutations encoded a T307I substitution in spike, linked with R682L at the furin cleavage site, and a T7I substitution in the M gene found both in isolation and linked with 2 different spike NTD mutations (Fig 2A). Overall, these results demonstrated that SARS-CoV-2 can accumulate considerable genetic diversity, as revealed by analysis of HT-SGS data at the single-genome level.

## HT-SGS performance in direct *ex vivo* sequencing of SARS-CoV-2

We anticipated that, compared to high-quality RNA preparations from cultured virus, human respiratory samples would contain variable levels of intact SARS-CoV-2 genomes, and that contaminants and inhibitors of steps in the HT-SGS process might also be present. We therefore evaluated the performance of HT-SGS using upper respiratory samples from 7 people with COVID-19 (S1 Table). Using droplet-digital reverse-transcription PCR (ddRT-PCR) to quantify two regions within the SARS-CoV-2 N gene, we detected virus loads in these samples ranging from 314 to >3 million RNA copies/mL. By comparison, our recovery of cDNA

**Table 1. Detection of input clonal sequences and recombinants in HT-SGS validation experiments.**

| Input wt:2M ratio | Count (%) of single-genome consensus seqs detected | | |
|---|---|---|---|
| | **wt** | **2M** | **Recombinant** |
| wt only | 84 (100) | 0 | 0 |
| 2M only | 0 | 162 (100) | 0 |
| 1:1 | 52 (37.7) | 86 (62.3) | 0 |
| 1:5 | 24 (13.6) | 153 (86.4) | 0 |
| 5:1 | 89 (84.8) | 16 (15.2) | 0 |
| 1:50 | 2 (1.2) | 162 (98.8) | 0 |
| 50:1 | 128 (97.7) | 3 (2.3) | 0 |

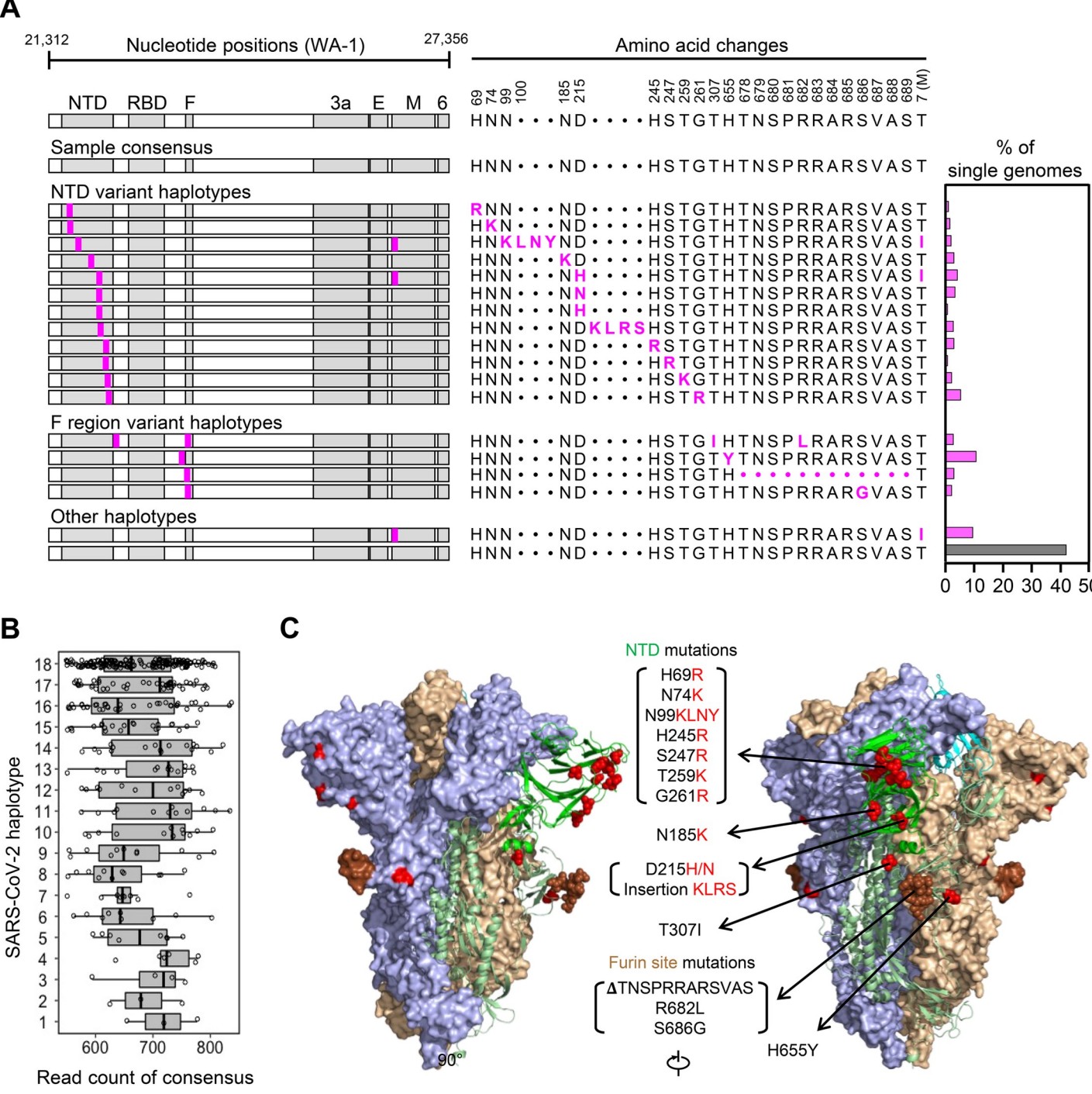

**Fig 2. Analysis of SARS-CoV-2 genetic diversity *in vitro*.** (A) Haplotype diagrams (left) depicting SARS-CoV-2 SGS detected in a 4th-passage Vero cell culture of the WA-1 reference clinical isolate. Spike NH₂-terminal domain (NTD), receptor-binding domain (RBD), and furin cleavage site (F) regions are shaded grey, with remaining regions of spike in white. Pink tick marks illustrate mutations relative to the sample consensus sequence. Amino acid changes corresponding to these mutations are shown in sequence alignment form (middle), with the percentage of all SGS in the sample matching each haplotype shown in the bar graph (right). The grey bar in the graph indicates the haplotype that matches the sample consensus sequence; variant haplotypes with at least 1 mismatch to sample consensus are in pink. (B) Read counts of each UMI bin for which the SARS-CoV-2 sequence matched each of 18 different haplotypes in Vero cell culture of the WA-1 clinical isolate. Bars indicate median read counts among bins. (C) Mapping of detected spike gene mutations on the trimer structure. Two protomers of the SARS-CoV-2 spike (PDB ID: 6zge) are shown in surface representation and colored light blue and wheat, respectively. The third protomer is shown in cartoon representation with the NTD region colored in bright green. NTD mutations as well as T307I and H655Y are shown in red and the furin cleavage site mutations are in brown. The molecular structures were prepared with PyMOL (https://pymol.org).

**Table 2. Virus loads and recoveries of cDNA and final SGS in HT-SGS from upper respiratory swab samples.**

| Sample | N1 RNA (copies/mL) | N2 RNA (copies/mL) | cDNA copies recovered[a] | Input cDNA copies (SGS) | SGS recovered | SGS % recovery |
|---|---|---|---|---|---|---|
| Pt.1 (d9) | 3,069,099 | 2,832,963 | 24,233 | 8,220 | 1,276 | 15.5 |
| Pt.1 (d11) | nd | | 19,576 | 10,000 | 882 | 8.8 |
| Pt.1 (d13) | 314 | 386 | 124 | 124 | 16 | 12.9 |
| Pt.1 (d15) | 13,470 | 11,105 | 1,807 | 1,807 | 284 | 15.7 |
| Pt.1 (d17) | 3,774 | 2,919 | 70 | 70 | 12 | 17.2 |
| Pt. 2 (d12) | 116,508 | 108,586 | 536 | 536 | 70 | 13.1 |
| Pt.3 (d17) | nd | | 17,531 | 10,000 | 1,210 | 12.1 |
| Pt. 4 (d8) | 872,984 | 841,366 | 605 | 605 | 108 | 17.9 |
| Pt. 5 (d8) | 2,669,500 | 2,520,722 | 4,060 | 3,400 | 367 | 10.8 |
| Pt. 6 (d8) | 105,735 | 92,156 | 255 | 255 | 31 | 12.2 |
| Pt. 7 (d16) | 101,327 | 96,916 | 50 | 50 | 13 | 26.0 |

[a]Sample volumes used for extraction were 140 μL ~ 300 μL.

encompassing the S, E, and M gene region in HT-SGS was considerably lower (Table 2). This discrepancy was consistent with multiple differences between the two measurements, including the presence of subgenomic RNAs containing ddRT-PCR target but lacking intact HT-SGS target sequences; lower efficiency of cDNA synthesis across our 6.1-kilobase HT-SGS target region than across short ddRT-PCR targets; and some degree of RNA degradation preferentially affecting HT-SGS. Similarly, SGS yields ranged from 8.8% to 26.0% of input cDNA copy numbers (Table 2), likely due to a combination of cDNA degradation and loss; failure of some cDNA molecules to amplify during PCR; and highly stringent read count cutoffs that we employed in the bioinformatic analysis in an effort to ensure accuracy of all reported sequences. Despite these considerations, however, yields at each step of the process were correlated with sample virus loads (S3 Fig), with recovery of between 12 and 1,276 single-genome consensus sequences for the samples studied (Table 2). Moreover, although we sequenced these samples to a high depth (7,499–462,919 raw reads/sample), we observed that detection of distinct virus haplotypes was highly reproducible in random subsamples down to a level of 5% (S4 Fig). This indicates that multiple samples can be combined in individual HT-SGS runs while still achieving sufficient sequencing depth for minor variant detection.

## Cross-sectional analysis of SARS-CoV-2 diversity and humoral immunity during acute COVID-19

Because the mutations we detected in cultured virus resembled those described for SARS-CoV-2 and other viruses during culture adaptation, we interpreted the extensive diversity observed as evidence of virus diversification *in vitro* rather than in the source patient. We therefore analyzed the diversity of HT-SGS sequences obtained from the 7 study participants in S1 Table. In samples taken between 8 and 17 days since the onset of clinical illness (each representing the earliest available sample for the individual), we detected only a single virus haplotype in participants 1, 2, and 6 (range of SGS counts, 31-1276/participant) and 2–3 haplotypes in each of the remaining 4 participants (range of SGS counts, 13-1210/participant; Fig 3). In addition, we noted no clear structural signature among the 7 mutations that defined intra-individual variant haplotypes, with 1 SNV in the downstream region of the spike gene, 4 SNVs in the non-structural ORF3 and ORF6 genes, and 2 synonymous SNVs (Fig 3). Overall, therefore, cross-sectional HT-SGS analysis of SARS-CoV-2 sequences in 7 individuals was notable for relative sequence homogeneity, as compared to results from cultured virus.

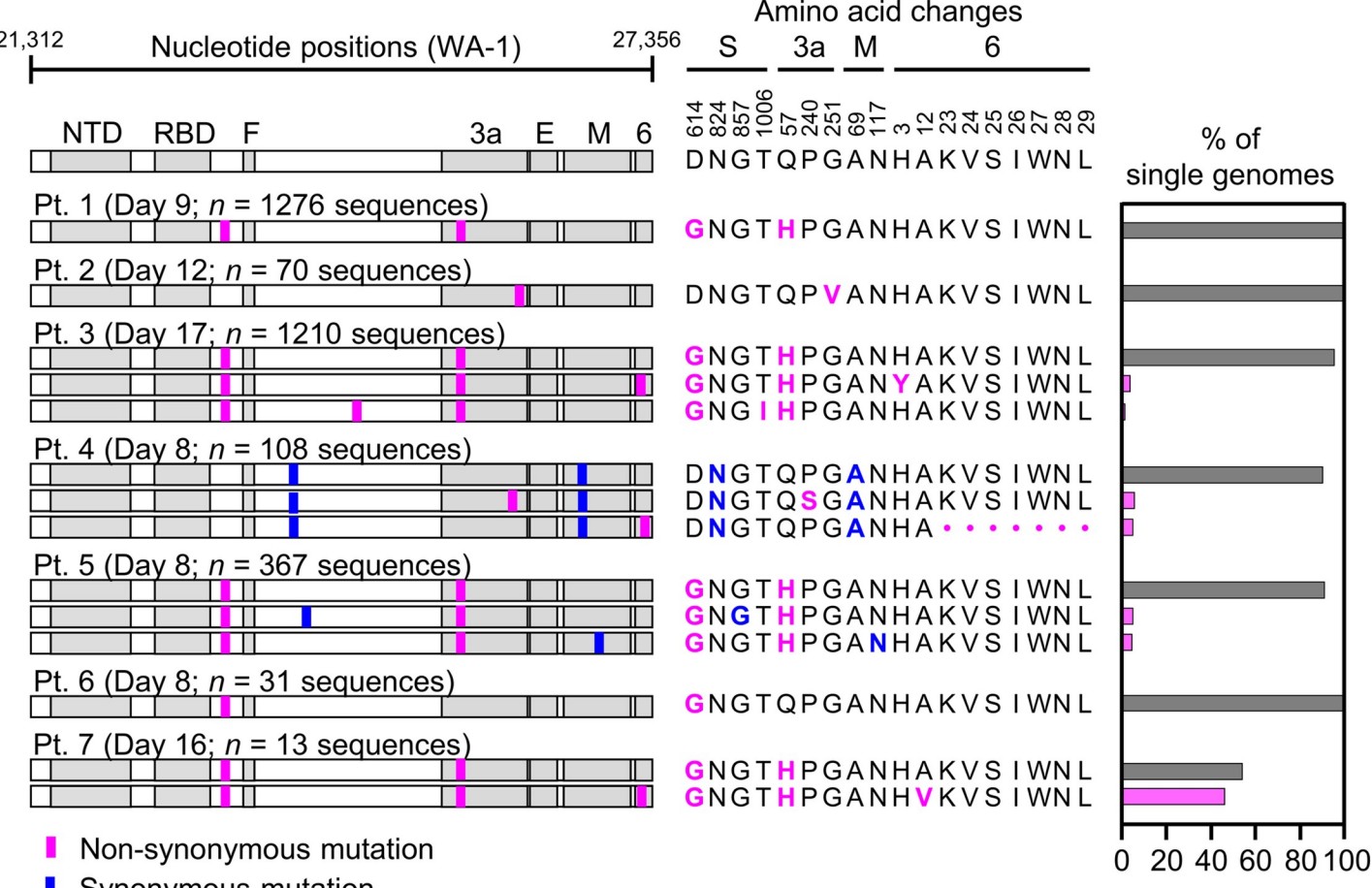

**Fig 3. Variant haplotypes of the SARS-CoV-2 virion surface protein gene region detected in upper respiratory tract samples from 7 hospitalized study participants with COVID-19.** Each participant label indicates day of clinical illness and the number of SGS obtained for the sample in parentheses. Haplotype diagrams (left) depicting SARS-CoV-2 SGS are as in Fig 2. Non-synonymous or synonymous mutations in each haplotype relative to the WA-1 reference sequence are shown with pink or blue tick marks. Amino acid changes (middle) and percentages of all SGS in the sample attributable to indicated haplotypes (right) are as in Fig 2. The haplotype matching the consensus for each sample is represented in grey; variant haplotypes with at least 1 non-synonymous mismatch to sample consensus are in pink.

To reconcile the extensive diversity among SARS-CoV-2 genomes *in vitro* with the lesser diversity detected in *ex vivo* samples, we hypothesized a relationship between virus diversity and host antibody responses arising after the establishment of infection. To investigate this, we used biolayer interferometry (BLI) to analyze antibody profiles in participants from whom longitudinal serum samples were available (i.e., participants 1 and 3). In these individuals, we observed a marked rise in autologous serum binding to spike protein between the earliest available timepoint (participant 1, day 9 and participant 3, day 17) and later timepoints (participant 1, days 16 and 19 and participant 3, day 27; Fig 4). The increase in total serum binding to spike was 6.2-fold between days 12 and 16 in participant 1 and 5.75-fold between days 17 and 27 in participant 3. Using monoclonal antibody (mAb) competition to map domain-specific responses, we detected serum binding to NTD, receptor-binding domain (RBD), and S2 domain in both participants (Fig 4). We also observed a continued increase in serum binding not competed by any tested mAb panel in participant 1 (Fig 4A, days 16 and 19, grey bars), suggesting progressive broadening of the binding response. Taken together, these findings

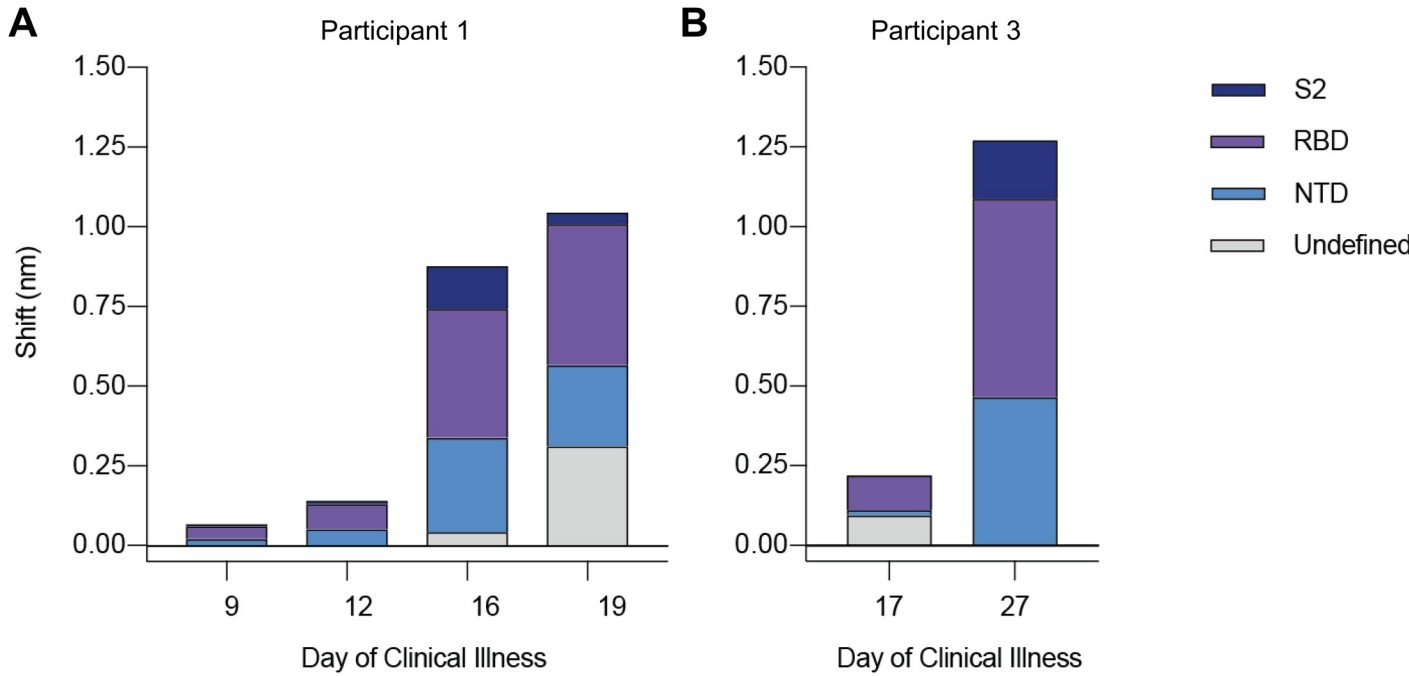

**Fig 4. Longitudinal analysis of participants 1 and 3 serum reactivity to binding domains of SARS-CoV-2 spike (S-2P).** (A and B) Reactivity to each domain was determined by preincubation of S-2P with competing mAbs targeting that domain before measuring serum binding using BLI. Total bar height indicates the binding response without competition and is reported at saturating timepoint. Stacked bars indicate proportions of binding attributable to S2 (dark blue), RBD (purple), and NTD (blue) regions, as inferred from relative reduction in total binding produced by mAb competition. Undefined (grey) stacked bars indicate proportions of total binding not competed by any mAb panel used. Plotted results represent averages of 2–4 replicate experiments for each condition.

indicated that samples with low levels of SARS-CoV-2 variation had been taken before full development of circulating antibody responses to the virus spike.

## Intra-individual SARS-CoV-2 evolution during acute infection

We next investigated the relationship between mounting spike-directed antibody responses and the levels and sequences of SARS-CoV-2 RNA in respiratory secretions from participant 1. We found that the burden of SARS-CoV-2 RNA declined substantially but irregularly between days 9 and 17 (Fig 5A). Between days 9 and 13, virus RNA declined by nearly 4 orders of magnitude, from $2.83 \times 10^6$ (N2)– $3.0 \times 10^6$ (N1) copies/mL to $3.14 \times 10^2$ (N1)– $3.86 \times 10^2$ (N2) copies/mL. However, virus RNA subsequently increased to $1.11 \times 10^4$ (N2)– $1.35 \times 10^4$ (N1) copies/mL on day 15, before declining again on day 17. This pattern was associated with the emergence of 2 minor variant SARS-CoV-2 haplotypes on day 11 and 4 minor variant haplotypes on day 15 (Fig 5B). Strikingly, these variants together bore 3 independent non-synonymous mutations within a single NTD epitope. On day 11, a C-to-T transition causing an H-to-Y change at amino acid residue 146 was found in 10/882 (1.1%) genomes sequenced. After a low virus RNA burden on day 13 with detection of only the consensus virus variant, sequencing on day 15 revealed deletions of either residues 141-144LGVY or residue 144Y alone. These mutations were found in 3 different haplotypes that accounted for 70/284 (26.1%) genomes sequenced on day 15 (Fig 5B, bar graph). Structural modeling onto the spike trimer (Fig 5C) indicated that these mutations were located in a supersite of vulnerability targeted by potent neutralizing antibody 4A8 [24], where similar mutations have been observed in case reports of persistent infections [5, 6] and a larger study of recurrently deleted regions [9]. Therefore, we performed additional serum antibody mapping studies with this mAb and found that before

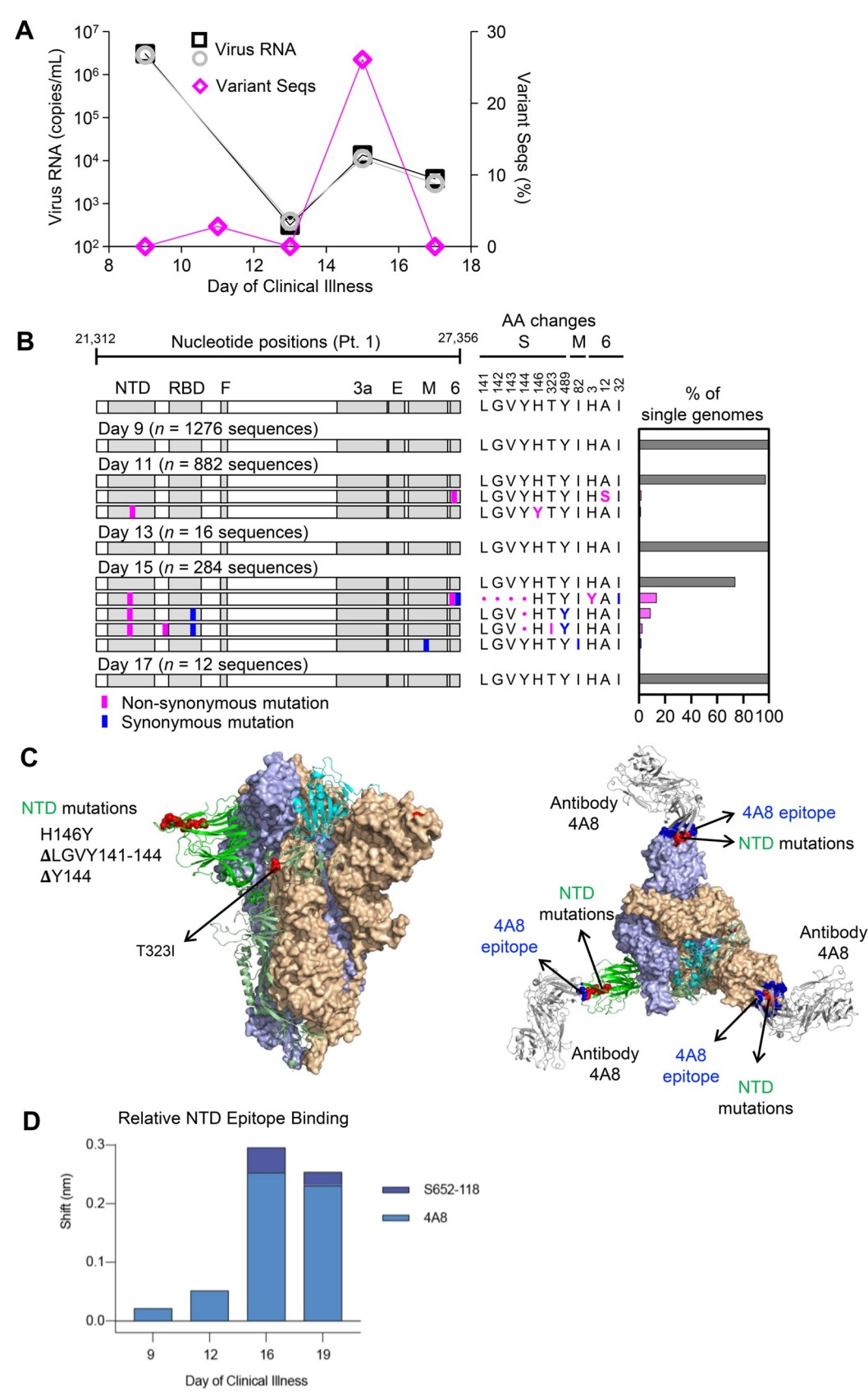

**Fig 5. Longitudinal analysis of SARS-CoV-2 RNA burden, SGS, and epitope-specific antibody binding to spike in participant 1.** (A) Copy numbers of SARS-CoV-2 N1 (black squares) and N2 (grey circles) RNA (left y-axis) and percentage of SGS not matching the predominant/consensus haplotype (pink diamonds, right y-axis) plotted for upper respiratory tract samples from days 9–17. (B) Variant haplotypes of the SARS-CoV-2 virion surface protein gene region detected on days 9, 11, 13, 15, and 17. The number of SGS obtained at each day is in parentheses. Haplotype diagrams (left), amino acid changes (middle), and percentages of all SGS in the sample attributable to indicated haplotypes (right) are as in Figs 2 and 3. The haplotype matching the consensus for each sample is represented in grey; variant haplotypes with at least 1 non-synonymous mismatch to sample consensus are in pink; one variant haplotype differing from sample consensus by only a synonymous mismatch is in blue. (C) Mapping of detected spike gene mutations on the trimer structure, viewed from the side (left) and top (right). The protomers in the spike (PDB ID: 6zge) were shown and colored with the same scheme as in Fig 2C. Detected mutations are highlighted in red. Antibody 4A8 (PDB ID: 7c21) is shown to bind to NTD with its epitope (blue) overlapping with the detected NTD mutations (right). The molecular structures were prepared with PyMOL (https://pymol.org). (D) Relative contribution of NTD epitope-specific serum antibodies to total NTD domain-specific binding on days 9, 12, 16, and 19. Plotted results represent averages of 2–4 replicate experiments for each condition.

the NTD mutations had emerged in autologous viruses, autologous serum antibodies against NTD predominantly recognized the 4A8 epitope (Fig 5D). Taken together, these results demonstrated a close temporal relationship between the development of SARS-CoV-2 spike NTD-specific antibodies in serum, the independent emergence of multiple mutations in a region of the NTD targeted by these antibodies, and a transient delay in virus clearance.

## Discussion

Here we developed and validated a novel method that accurately sequences the 6.1-kilobase SARS-CoV-2 surface protein gene region from large numbers of individual virus genomes. Using this method, we analyzed virus genetic diversity both *in vitro* and in respiratory secretions from people with COVID-19. In contrast to *in vitro* passaged viruses, which exhibited extensive diversity fitting patterns associated with culture adaptation [21–23], we initially found relatively low intra-individual SARS-CoV-2 diversity *ex vivo*. These results appeared consistent with the slow evolution among worldwide virus sequences during the early months of the pandemic [1]. Nevertheless, our relatively homogeneous cross-sectional sequencing findings in people with COVID-19 were not due entirely to intrinsic limitations on SARS-CoV-2 diversity. Instead, longitudinal analysis during the second and early third weeks of illness in one person revealed a transient increase in virus burden and multiple new virus variants in which 3 different mutations in an epitope of the spike NTD had arisen independently. The mutated epitope was previously shown to be a neutralizing antibody target [24], and was identified herein as a major target for antibodies in the autologous serum. Our results therefore suggest selection of SARS-CoV-2 spike variants by mounting antibody responses in the acute setting.

Mutational evasion of adaptive immune responses by SARS-CoV-2 during acute COVID-19 has not been clearly documented previously. This relationship may have been overlooked in part due to the emphasis on tracking new mutations on a global scale, with a predominance of cross-sectional rather than longitudinal analyses of infected individuals. The early peak of SARS-CoV-2 RNA in respiratory secretions may also favor high-quality data acquisition in very early infection, leading to overrepresentation of individuals in whom virus populations have not yet been subjected to adaptive immune pressure. Another important consideration is the sequencing method used. Our method was specifically developed for high-throughput analysis of single virus RNA molecules, and incorporates several layers of error correction that aid in distinguishing true variation from technical errors. This allowed groups of important virus variants to be detected even though each variant individually accounted for a small

proportion of all sequences in each sample. Finally, we cannot rule out that our distinctive findings might relate to our longitudinal study participant's history of stem cell transplantation. It is possible that immune suppression can lead to higher levels of virus replication and thus an unusually rapid accumulation of "total-body" virus diversity *in vivo*. However, we noted that our longitudinal participant was no longer receiving immune suppressive medication at the time of COVID-19 diagnosis, and measurements of virus burden in respiratory secretions were consistent with previous studies in immunocompetent participants [4, 25]. Wider application of our combined virological and immunological approach in diverse clinical cohorts will aid in defining circumstances under which SARS-CoV-2 genetic variants may emerge under immune-mediated pressure.

It is important to emphasize that our bioinformatic strategy in this study prioritized the accuracy of reported variants over attempts to interpret very rare, potentially spurious sequences. Because the sensitivity of SGS depends in part on the number of genomes analyzed in each sample, future HT-SGS studies can achieve higher sensitivity for uncommon variants by preparing larger quantities of sample, by further optimizing virus RNA yield, integrity, and downstream processing efficiency, and by analyzing anatomic sites outside the upper respiratory tract that could harbor genetically distinct virus subpopulations [13, 26]. More fundamentally, however, rare variant detection in our HT-SGS process is limited by the intrinsic error rate of the RT step. Based on a frequency estimate of approximately 0.00007 error/base [27], we anticipated that RT errors would introduce at least one incorrect base into every 2–3 sequences from our chosen target region in this study. These considerations motivated our use of variant calling, rare mutation reversion, and unique SGS exclusion, which together were sufficient to exclude low-frequency technical errors in validation studies while preserving key variants from *in vitro* and *ex vivo* virus populations. Nevertheless, this strategy also excludes any real mutations occurring at or below the RT error rate threshold, and thus does not reliably preserve randomly-generated variants that have not undergone a degree of selective expansion. Methodological refinements or new approaches that address this challenge may lead to important insights about the virus mutation rate in future studies.

Tracking intra-individual virus evolution is of great interest in understanding SARS-CoV-2 pathogenesis and treatment. As our longitudinal study participant recovered clinically, spike variants detected by HT-SGS were replaced by unmutated sequences even though the variant sequences might have avoided neutralization by 4A8-like antibodies *in vivo*. This was likely due to a broadly-targeted antiviral response, including innate defenses, antiviral T cells, and multiple antibody specificities, each potentially with distinct kinetics during the transition from acute infection to convalescence. The absence of spike RBD variants in our longitudinal sequencing despite strong RBD-directed serum binding suggests limitations on SARS-CoV-2 escape from polyclonal responses, perhaps especially in genome regions less tolerant of indel mutations [9]. Nevertheless, recent findings made with spike variants from second wave pandemic spread demonstrate that SARS-CoV-2 can sometimes overcome genetic barriers to broader immune escape [28–31]. At the same time, the diversity of clinical outcomes in COVID-19 may relate in part to control of the virus, with slower virologic clearance linked to disease severity [25, 32–34]. It will be important to examine whether this reflects a "tipping point" in early infection at which SARS-CoV-2 genetic diversity can occasionally allow sustained replication through the evasion of immune recognition. Immunity induced by prior infection, vaccination, or passive immunization could reduce the potential for escape by controlling initial levels of virus replication quickly. Our results also emphasize that early antiviral therapy or combinations of antivirals with distinct targets could have markedly higher virologic efficacy than monotherapy administered later in the disease course.

## Materials and methods

### Ethics statement

Individuals admitted as hospital inpatients at the U.S. National Institutes of Health (NIH) Clinical Center who had positive tests for SARS-CoV-2 were enrolled consecutively for combined virological and immunological analysis during the period of March-May 2020 (S1 Table). Study participants were recruited in compliance with relevant ethical regulations and provided informed consent under the AMOEBAE protocol (#10-I-0197) or the ACTT-1 protocol (#20–0006), which were approved by the NIH Institutional Review Board and the ADVARRA Institutional Review Board, respectively.

### Samples

Plasmid DNA for validation experiments was generated by BioInnovatise, Inc. (Rockville, MD) to include the WA-1 sequence (GenBank–MN985325) of the 6.3-kilobase region containing the S, ORF3, E, and M genes, inserted into the pSI vector (Promega). A double-mutant plasmid was then created by using site-directed mutagenesis to scramble 20 bases each at the 5' and 3' ends of the target (genome position 21,583 –ATTGCCACTAGTCTCTAGTC ➔ CCCTAATTGTTGAATCGCCT and genome position 27,169 –ATATTGCTTTGCTTGTAC AG ➔ TCTGGTTGAGCTACTATTTA; Fig 1B). To prepare clonal RNA samples representing these two sequences, plasmids were linearized by digestion with AatII (FD0994, ThermoFisher Scientific) and *in vitro* transcribed using the MegaScript T7 Transcription Kit (AMB1334, ThermoFisher Scientific). Reactions were incubated at 4˚C for 20 hr to minimize incomplete transcripts [35]. Plasmid DNA was then removed using the TURBO DNA-free kit (AM1907, ThermoFisher Scientific), and RNA was recovered by lithium chloride precipitation. The RNA was quantified on a Qubit Fluorometer and Quant-iT RNA assay kit (Q10213, Thermofisher Scientific) and analyzed by electrophoresis with E-Gel EX Agarose Gels 1% (G401001, Thermofisher Scientific).

Extracted RNA from the 4[th] Vero cell passage of the SARS-CoV-2 WA-1 clinical isolate was obtained from the BEI Resource (catalog #NR-52285). Nasopharygneal or oropharyngeal swabs from study participants were collected in viral transport medium and cryopreserved until processing.

### SARS-CoV-2 RNA quantification

Total RNA was extracted from oropharyngeal and nasopharyngeal swab specimens using the QIAamp Viral RNA Mini Kit (Qiagen, Germantown, MD, USA) according to the manufacturer's protocols. The QX200 AutoDG Droplet Digital PCR System (Bio-Rad, Hercules, CA, USA) was used to detect and quantify SARS-CoV-2 RNA using the SARS-CoV-2 Droplet Digital PCR Kit (Bio-Rad), which contains a triplex assay of primers/probes aligned to the CDC markers for SARS-CoV-2 N1 and N2 genes and human RPP30 gene. 96-well plates were prepared with technical replicates containing 5.5 µL RNA per well. Microdroplet generation was performed on the QX200 Automated Droplet Generator (Bio-Rad), and plates were sealed with the PX1 PCR Plate Sealer (Bio-Rad) before proceeding with RT-PCR on the C1000 Touch Thermal Cycler (Bio-Rad) according to the manufacturer's instructions. Plates were read on the QX200 Droplet Reader (Bio-Rad) and analyzed using the freely available QuantaSoft Analysis Pro Software (Bio-Rad) to quantify copies of N1, N2, and RPP30 genes per well, which was then normalized to mL of sample input.

## HT-SGS sample preparation and sequencing

Nasopharyngeal or oropharyngeal swab fluids were thawed and centrifuged at 1,150 x *g* for 15 min at room temperature to pellet cells and debris. Supernatants were transferred to separate tubes, and supernatant and pellet fractions were processed in parallel, although SGS derived from these two fractions were subsequently found to be similar and were thus pooled for each sample in the final analysis. Nucleic acids were extracted from supernatants and pellets using the QIAamp Viral RNA Mini Kit (52906, Qiagen) according to the manufacturer's instructions.

Sample RNA was reverse transcribed with SuperScript IV Reverse Transcriptase (18090010, ThermoFisher Scientific) using an RT primer binding within the SARS-CoV-2 ORF6 gene (TCTCCATTGGTTGCTCTTCATCT, WA-1 reference positions 27,357–27,379). The RT primer also included an 8-base UMI (NNNNNNNN) and an outer reverse primer binding site for PCR amplification (CCGCTCCGTCCGACGACTCACTATA; see S1 Fig and S1 Table). Virus cDNA was treated with proteinase K for 25 min at 55˚C with continuous shaking to remove residual protein [36], followed by purification with a 2.2:1 volumetric ratio of RNA-Clean XP solid phase reverse immobilization (SPRI) beads (A63987, Beckman Coulter). Copy numbers of resulting cDNAs were determined by limiting-dilution PCR using fluorescence-assisted clonal amplification (FCA) [37] and a gene-specific primer pair detecting a region upstream of the S gene (S2 Table). Subsequently, cDNA molecules were amplified using the Advantage 2 PCR kit (639206, Takara Bio) with initial denaturation at 95˚C for 1 min, followed by 30 cycles of denaturation at 95˚C for 10 sec, annealing at 64˚C for 30 sec, and extension at 68˚C for 7 min, followed by one final extension at 68˚C for 10 min. Each PCR reaction was run in a 20 μL volume with final primer concentration of 400 nM. Primers included the outer reverse primer and one of two different forward primers (S2 Table). Amplified DNA was quantified on a Qubit Fluorometer (Thermofisher Scientific) and analyzed by electrophoresis with precast 1% agarose gel (Embi Tec) or the Agilent High Sensitivity DNA kit (5067–4626, Agilent). Amplified DNA products spanning the 6.1-kilobase virion surface protein gene region of SARS-CoV-2 with single-genome UMI-based tagging were incorporated into sequencing libraries using the SMRTbell Express Template Prep Kit 2.0 (100-938-900, Pacific Biosciences) and Barcoded Overhang Adapters (101-629-000, Pacific Biosciences) to enable sample multiplexing. Libraries were prepared for sequencing by primer annealing and polymerase binding using the Sequel II Binding Kit 2.0 and Int Ctrl 1.0 (101-842-900, Pacific Biosciences), and were sequenced by single-molecule, real-time (SMRT) sequencing using a Sequel II system 2.0 (Pacific Biosciences) with a 30-hour movie time under circular consensus sequencing (CCS) mode.

## HT-SGS initial data processing

Circular consensus sequences (CCS) were generated from SMRT sequencing data with minimum predicted accuracy of 0.99 and minimum number of passes of 3 in Pacific Biosciences SMRT Link (v8.0) using Arrow modeling framework [38]. CCS reads were then demultiplexed using Pacific Biosciences barcode demultiplexer (lima) to identify barcode sequences. The resulting FASTA files were reoriented into 5'-3' direction using the *usearch -orient* command in USEARCH (v8.1.1861) [39]. Cutadapt (v2.7) [40] was used to trim forward and reverse primers. Length filtering was performed to remove reads shorter than 90% or longer than 130% of the reference sequence length. Appropriately-sized reads were then binned using 8-base UMI sequences. The read count in each UMI bin was plotted against the rank of that UMI bin (on log scale) within the sample, and the inflection point (i.e., point of concavity change) was calculated (S2B Fig). UMI bins with read counts lower than the inflection point

were discarded, leaving UMI bins with higher counts. Cutadapt (v2.7) was used to remove the RT primer and UMI sequences from each UMI bin consensus to obtain the SARS-CoV-2 insert sequence for that bin. Consensus sequences were generated for each bin using the *usearch-cluster_fast* command based on 99% identity to obtain high-confidence single-molecule sequences. Consensus sequences were then analyzed by searching the BLAST nt database, and non-coronavirus sequences thus identified were discarded.

## Determining SGS in HT-SGS data

The probability that two independent UMI sequences differ by a single nucleotide substitution (i.e., have an edit distance of 1 base) can be estimated using binomial distribution with parameters $n = 8$ and $p = 0.75$, where $n$ is the number of independent UMI bases and $p$ is the probability that a base differs between two UMIs. Therefore, the probability of any two independent UMIs having edit distance one is $B(8, .75, 1) = 3.6E-4$. Hence, it is appropriate to assume that two UMI sequences having edit distance 1 could represent a scenario where one of the UMIs is derived from the other through PCR and/or sequencing error. To identify and remove potential false UMI bins, we utilized a UMI network method [41]. In this network, each UMI sequence is represented by a node. Given two distinct nodes $a$ and $b$ with read counts $n_a$ and $n_b$, respectively (assume $n_a \geq n_b$), $a$ and $b$ are connected by an edge if they have edit distance 1 and satisfy the following count criterion: $n_a \geq 2n_b - 1$. To resolve the network formed above, we applied the *adjacency* method [41]. According to this method, the node with the largest count was selected and all connected nodes were removed. Next, the node with the second largest count was selected and all connected nodes were removed. This process was repeated until no more edges remained in the network. The *adjacency* method allowed resolution of a complex network to a single node. To further reduce the likelihood of including false UMI bins in downstream analysis, we combined our network adjacency approach with a knee point (i.e., point of maximum curvature) filter (S2C Fig) to ensure that UMIs with large total counts were preserved. Inflection and knee points can both be considered as separations between the high- and low-count UMI bins, and both depend on the shape of the count distribution. The knee point is more conservative in comparison to the inflection point. We used the knee rather than the inflection point at this stage in order to provide a more stringent threshold for removing false bins. To identify virus haplotypes defined by the data, we took the consensus sequences of all UMI bins and collapsed non-unique sequences. We considered the unique sequences bearing different combination of mutations as individual haplotypes. Finally, we manually inspected alignments of remaining UMI bin consensus sequences and removed any sequence that represented a SARS-CoV-2 haplotype observed in only one UMI bin for the sample in which it was found.

## UMI collision estimates

We investigated the possibility of UMI collision (two distinct molecules labeled with the same UMI) based on the assumption of uniformly distributed UMIs. As described by Fu et al. [42], the expected number of unique UMIs captured is k = $m[1 - e^{-n/m}]$, where n is the number of molecules and m is the size of UMI pool. Therefore, $n = -m \ln\left(1 - \frac{k}{m}\right)$. Given the number of observed unique UMIs in a particular sample and UMI pool of size $4^8 \approx 65000$, we estimated the number of molecules and calculated the number of UMI collisions, (n-k), for each sample. This number was observed to be small, and the probability of collision in each sample was at most 4%, with an average 1.8% across all samples. We also note that, in the event of a UMI collision between two distinct sequences, the clustering and consensus formation for each UMI

bin described above and in S2 Fig results in preservation of the sequence cluster with higher read abundance and removal of the sequence cluster with lower read abundance.

## Variant calling

Despite high single molecule read accuracy (>99.9%) of Pacific Bioscience HiFi reads, some sequencing errors–particularly small indels–may persist in the reads after applying CCS read correction. These errors and those that may arise during conversion of RNA to cDNA may not be identified by our extensive UMI-based error correction method. To distinguish such errors from real biological variation, we used 'Map Long reads to reference' tool in 'Long read support' plugin in the CLC Genomics Workbench v.20.0.4 (GWB) with default settings. This tool utilizes Minimap2 to map long reads [43]. We used the WA-1 reference sequence (GenBank accession: MN985325.1) as a reference during mapping. We employed the Low Frequency variant caller in the GWB with the following settings:

$$Ignore\ broken\ pairs = None$$

$$Minimum\ coverage = 5$$

$$Minimum\ count = 4$$

$$Minimum\ frequency\ (\%) = 0.0$$

We also applied a filtering criterion to remove variants in homopolymer regions with minimum length of 2 and a frequency less than or equal to 20%. We did not consider quality or direction and position filters typically used in analyzing paired-end, short-read data as these do not apply to long-read amplicon sequencing. We then manually inspected the mutation list to remove presumptive artifacts that were missed by the variant callers. The positions identified in our high-confidence variants list were then masked in the read mapping and bases in all other positions were reverted to the reference base, where applicable, using an in-house python script.

## Analysis of serum antibody binding to SARS-CoV-2 spike protein

Domain-specific antibody competition assays using a His-tagged SARS-CoV-2 Spike protein ectodomain containing 2 proline stabilization mutations (S-2P) [44] were performed using a fortéBio Octet HTX instrument and His1K (anti-penta His) biosensors at 30˚C with agitation set to 1,000 rpm. Biosensors were first equilibrated for 600 seconds in PBS supplemented with 1% BSA, 0.01% Tween-20, and 0.02% sodium azide (PBS-BSA). Purified S-2P (10 μg/mL in PBS-BSA) was immobilized on equilibrated His1K sensors for 600 s. S-2P protein loading onto to the sensors was between 0.9 and 1.3 nm shift. Following S-2P immobilization, biosensors were equilibrated in PBS-BSA for 60 s. S-2P coated biosensors were submerged in either S-2P binding-domain specific competitor monoclonal antibodies (mAb) or negative control antibody, each at 10 μg/mL in PBS-BSA, for 600 s. At 600 s, the binding of all S-2P binding antibodies was saturating. Competitor mAbs were divided into three separate groups, each targeting a binding domain of S-2P: RBD, NTD, and S2 domain. Monoclonal antibodies included were composed of human IgG RBD-specific antibodies LY-CoV-555 [45], S309 [46], CR3022 [47], and CB6 [48], NTD-specific antibodies S652-118 [49], 4–8 [50] and 4A8 [24] and S2-specific antibody S652-112 [49]. Following saturating competitor mAb association, biosensors were equilibrated in PBS-BSA for 60 s and then submerged in serum samples diluted 100-fold in PBS-BSA for 3600 s. Raw sensorgrams datapoints were aligned to Y (nm) = 0 in at the beginning of the second association phase. Competition and serum shift were analyzed when

the serum samples reached saturation (4001.2 s). Pie charts depict each binding domain's relative contribution to the overall serum antibody binding to S-2P, as determined by percent competition. Percent competition (% C) of serum antibody binding to S-2P by competitor mAb groups was calculated using the following formula: % C = [1 −(shift nm value at 4001.2 s in presence of competitor mAb)/(shift nm value at 4001.2 s in presence of negative control antibody)]*100. All assays were performed in duplicate.

## Supporting information

**S1 Fig. Details of HT-SGS process from sample to sequencing.** SARS-CoV-2 genomic RNA (gRNA) is reverse-transcribed with a primer that binds in ORF6, downstream of the M gene stop codon, and includes a UMI sequence of 8 random nucleotides flanked by a PCR reverse primer binding site. Reverse-transcription products are amplified by PCR using a forward primer that binds in ORF1, upstream of the spike gene start codon. Amplified products are then subjected to long-read sequencing.
(PDF)

**S2 Fig. Details of HT-SGS data analysis.** (A) Bioinformatic pipeline, depicting sequential workflow steps and tools used. Black boxes show tasks at each step, with the tools used in the grey boxes, and the outputs in the blue bubbles. (B) Initial exclusion of false UMI bins based on read count distribution on a log scale. The dashed line indicates the read count inflection point below which UMI bins in this sample were excluded. (C) Final exclusion of low count UMI bins based on read count distribution on a log scale. The dashed line indicates the read count knee point below which UMI bins in this sample were excluded, following initial false bin removal from the sample and network adjacency. Data are presented for the cultured virus sample presented in Fig 2.
(PDF)

**S3 Fig. Relationships between inputs and yields of steps in the HT-SGS data generation process.** (A) Comparison of virus load of original sample with total cDNA synthesis yield. (B) Comparison of cDNA input copies from each sample with final SGS counts.
(PDF)

**S4 Fig. Effect of downsampling on haplotype detection.** Each subsample was generated by random draws of a fixed percentage from reads without replacement. This process was repeated 100 times for each percentage. (A) The initial numbers of UMI bins (y-axis) are shown for different degrees of downsampling (x-axis). (B) The minimum read counts per UMI bin (y-axis) are shown for different degrees of downsampling (x-axis). (C) Proportion of each haplotype present in the 100% sample and in each subsample. Data analyzed are from sequencing of participant 1, day 15.
(PDF)

**S1 Table. Clinical characteristics of study participants.**
(PDF)

**S2 Table. Primer sequences used in HT-SGS procedures for this study.**
(PDF)

## Acknowledgments

We gratefully acknowledge the participants in this study and thank the Vaccine Research Center Genome Analysis Core for sequencer access.

## Author Contributions

**Conceptualization:** Sung Hee Ko, Elham Bayat Mokhtari, Prakriti Mudvari, Eli A. Boritz.

**Formal analysis:** Elham Bayat Mokhtari, Prakriti Mudvari.

**Funding acquisition:** Peter D. Kwong, Daniel S. Chertow, Nancy J. Sullivan, Eli A. Boritz.

**Investigation:** Sung Hee Ko, Elham Bayat Mokhtari, Prakriti Mudvari, Sydney Stein, Christopher D. Stringham, Danielle Wagner, Sabrina Ramelli, Marcos J. Ramos-Benitez, Tongqing Zhou, John Misasi.

**Resources:** Jeffrey R. Strich, Richard T. Davey, Jr., Daniel S. Chertow.

**Supervision:** Peter D. Kwong, Daniel S. Chertow, Nancy J. Sullivan, Eli A. Boritz.

**Writing – original draft:** Sung Hee Ko, Elham Bayat Mokhtari, Prakriti Mudvari, Eli A. Boritz.

**Writing – review & editing:** Sung Hee Ko, Elham Bayat Mokhtari, Prakriti Mudvari, Christopher D. Stringham, Danielle Wagner, John Misasi, Nancy J. Sullivan, Eli A. Boritz.

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
