## [Decision Letter · Decision Letter 0]

28 Jan 2021

Dear Dr. Boritz,

Thank you very much for submitting your manuscript "High-Throughput, Single-Copy Sequencing Reveals SARS-CoV-2 Spike Variants Coincident with Mounting Humoral Immunity during Acute COVID-19" for consideration at PLOS Pathogens. As with all papers reviewed by the journal, your manuscript was reviewed by members of the editorial board and by several independent reviewers. In light of the reviews (below this email), we would like to invite the resubmission of a significantly-revised version that takes into account the reviewers' comments.

Thank you for submitting to PLOS Pathogens. This is a nice piece of work that both reviewers appreciated. Both reviewers have constructive criticisms, that if incorporated would make for a much improved manuscript. Reviewer 1 in particular details a few experiments to get at the issue of method accuracy. Addressing this in some way is important for this promising method and interesting results.

We cannot make any decision about publication until we have seen the revised manuscript and your response to the reviewers' comments. Your revised manuscript is also likely to be sent to reviewers for further evaluation.

Sincerely,

Adam S. Lauring

Section Editor

PLOS Pathogens

Adam Lauring

Section Editor

PLOS Pathogens

Kasturi Haldar

Editor-in-Chief

PLOS Pathogens

orcid.org/0000-0001-5065-158X

Michael Malim

Editor-in-Chief

PLOS Pathogens

orcid.org/0000-0002-7699-2064

Thank you for submitting to PLOS Pathogens. This is a nice piece of work that both reviewers appreciated. Both reviewers have constructive criticisms, that if incorporated would make for a much improved manuscript. Reviewer 1 in particular details a few experiments to get at the issue of method accuracy. Addressing this in some way is important for this promising method and interesting results.

Reviewer's Responses to Questions

**Part I - Summary**

Reviewer #1: Ko et al. develop a new method for sequencing single virus genomes with long-read technology and apply it to a cell culture adapted virus and to 7 clinical SARS-CoV2 samples. The sequencing method developed in this manuscript seems quite useful, with the clear benefit of enabling haplotype resolution and mutation linkage. The authors also present an interesting case of within-host evolution that is correlated to antibody development. This is of particular interest now and provides a useful contribution to the discussion of the role of immune-compromised hosts to SARS-CoV2 evolution. Overall, I think that the authors conclusions and findings are reasonable and interesting, and that the method could be quite useful. However, some of the results are currently difficult to interpret without some additional controls to validate the sequencing method for its robustness across viral RNA input copies and total number of output single consensus genomes. I suggest some specific controls that I think would be useful for clarifying these points, as well as some points in the Methods that could be further clarified.

Reviewer #2: Ko et al. provide a timely report about sequence diversity in the Spike gene of SARS-CoV-2. They analyze both virus that has been grown in tissue culture and virus isolated from the respiratory tract from infected people. There is a critical need to follow Spike protein evolution both within and between people as this could lead to new strains that could escape the anticipated protection that will be provided by the current vaccines. The authors improve on existing sequencing approaches by using an SGS approach but at higher per-specimen sampling than is typically done, and this is done over a relatively large region of the genome (wedding the deep part of "deep sequencing" with SGS accuracy for sampling individual genomes). What has seemed like an obvious approach of using a UMI with PacBio long read technology has been difficult due to the high error rate of PacBio which makes it hard to accurately identify the UMI. Overcoming this problem will be a welcome addition to the field. The number of long reads one can get from a sample is limited by the number of long cDNA molecules that RT can generate (i.e. the number of templates), in this case 6.1 kb. The relatively high titers of SARS-CoV-2 samples conveniently overcomes this limitation by providing a larger number of starting RNA templates. The paper does an elegant job of linking the antibody response to longitudinal changes in the Spike protein sequence and antibody escape in one case but a case that is very informative.

This is an important piece of work. I have only a few technical questions. The biology in the paper is on mark addressing important questions.

**Part II – Major Issues: Key Experiments Required for Acceptance**

Reviewer #1: Major comments:

1. The authors document very reasonable and extensive bioinformatic controls for sequence quality, UMI consolidation, and consensus genome calling. However, the authors have not directly performed validation experiments that benchmark a. this method’s intrinsic error rate on a sample with no variation or b. its ability to correctly characterize within-host haplotype frequencies across various RNA input quantities and read counts. In my opinion, the authors need to perform some additional validation experiments to address these concerns. To address point a, the authors should perform a simple validation experiment in which the authors sequence a clonal RNA transcript and report the number of detected haplotypes and their frequencies. This would provide a raw error rate of their method, and provide a more direct way of measuring the error rate of their RT and PCR protocols than by relying on the variant calling and error correction method described in the Methods. Suggestions for addressing point b are below in comments 2 and 3.

2. Given the known sensitivity of within-host variant detection to input RNA copies, the authors should report in a table somewhere in the main text the number of RNA copies in each sample that is sequenced.

3. In the authors sequencing results, there is a wide range of the number of total single-genome consensus sequences generated for each sample. While in some samples the number is reasonably high >1000), in patients 6 and 7 and patient 1 on days 13 and 17, the total number of single genomes is quite low. The sequencing results from these samples are quite difficult to interpret as is. It seems to me that greater single genome consensus sequences should provide higher resolution of within-host haplotype frequencies, and I would guess that samples with higher viral RNA contents generally produce more single genome consensus sequences. However, these points are not made explicit and the authors have not designated a cutoff for either a viral load necessary for this method to be successful or for a number of successfully sequenced single genome consensus sequences for this method to accurately represent the within-host diversity of the sample. These values will be quite important for readers who may be interested in applying this new method, and are also important for interpreting the results shown in the paper. I would suggest that following revisions and control experiments, or something that accomplishes similar goals:

a. The authors should discuss in the text whether the number of single genome consensus sequences is correlated to the viral load of the original sample. A plot showing this relationship would be helpful. This is particularly important because the authors remark throughout the paper that their method can be used to characterize thousands of individual viral genomes, but they only ever report, at maximum, ~1200 single genomes for a given sample.

b. The authors could perform some sort of serial dilution experiment in which a known variant haplotype is spiked into a wild type, clonal RNA at some frequency, and then serial dilutions are generated and sequenced. This would provide important information about how to interpret haplotype frequencies derived from low copy number samples. I realize that this is bit more work, but it would provide information that would be quite useful. Alternatively, if the authors can perform some sort of experiment that clarifies the relationship between viral load and Method accuracy, that would be great.

c. The authors should assess in some way how the total number of unique single genome consensus sequences impacts haplotype detection. This could be accomplished bioinformatically by randomly subsampling down one of their samples with >1000 single genome consensus sequences to varying degrees, and comparing the output. This sort of information would be really helpful for interpreting the haplotype results for the samples with very low numbers of single genome consensus sequences.

While all three of these experiments may not be absolutely necessary, I do think that these questions should be addressed in some way in the revised version of the manuscript.

Reviewer #2: None

**Part III – Minor Issues: Editorial and Data Presentation Modifications**

Reviewer #1: Minor comments and points of clarification:

1. In the Methods, the authors report using 1000 cDNA input copies into the PCR reaction. This point was not clear to me. The authors are able to generate >1000 unique single genome consensus sequences for some samples, despite only using 1000 cDNA input copies. How is this possible? Is this just pipetting error? Why did the authors choose to limit their input to 1000 cDNA copies? The authors state multiple times in the manuscript that their method is designed to sequence thousands of individual viral genomes, but it seems that by limiting the copy numbers that they are by definition restricting their sequencing to ~1000 copies. The authors should clarify this in the text.

2. If possible, I would find it very helpful to see a calculation of: given the number of input RNA molecules, what is the probability that 2 different RNAs get labelled with the same UMI? Alternatively, how much in excess is the UMI vs the RNA? Some sense for this would be helpful.

3. On lines 211-216, the authors discuss the amount of virus in patient 1 as comparable to other, acutely infected patients, but do not discuss the duration of shedding. While the viral load is certainly important for within-host viral evolution, the duration of shedding is also critical given that it determines whether within-host viral replication is still occurring at the time of antibody development. The authors should acknowledge this and perhaps could cite some literature describing within-host evolution during prolonged infections.

4. The description of the node collapsing and adjacency method could be clarified a bit. It sounds to me like what is happening is that you connect UMIs that are separated by 1 and meet this number criteria and then you collapse them all into 1. Is that right? It is also not immediately clear to me what the difference between the knee point and inflection point is. The authors could add a sentence in the Methods to describe this for readers who may not be familiar with this.

5. It is not clear to me exactly when the variant calling correction is implemented. Is this done on the consensus sequences of each UMI bin? Or is this done on the raw reads themselves?

6. On line 263: “Supernatants were transferred to separate tubes, and supernatant and pellet fractions were processed in parallel, although single-genome sequences derived from these two fractions were subsequently found to be similar and were thus pooled for each sample in the final analysis.” Do the authors mean that the pellets and supernatants were combined for one particular set of samples, or that the reads were combined later? Were pellets and supernatants initially separated to compare within-host variants in genomic RNA vs. cell-associated RNA?

7. On line 304, what is the “log rank of that UMI bin within the sample”? I had assumed initially that the log rank was just an ordered list of UMIs, ordered by read count, but was unsure. Also, is Figure S2 showing the results for the WA-1 BEI sample or some other sample?

Reviewer #2: 1. The genomes appear to contain mostly single nucleotide polymorphisms with but maybe some examples of polymorphisms in two different genes. To get at the issue of recombination during PCR (which always happens) it would be good to be clear whether or not the linkage is perfect in the sequenced genomes or whether there has been any scrambling of the markers between genomes. Scrambling could have occurred in vivo or during PCR, but an absence of scrambling would indicate that it didn't happen during PCR. It still happens during PCR but the ability to build a consensus sequence for each template masks those events.

2. Since cDNA synthesis has poor processivity it would be interesting to know how many genome sequences are obtained as a function of copies of RNA used in the initial cDNA reaction, i.e. what is the efficiency of template utilization in this sequencing approach?

3. RT has a reported incorporation error rate of 1 in 10,000 nucleotides during cDNA synthesis (I believe listed in the manufacturers information). This would mean that every second genome would have an RT error incorporated. In this scenario a pool of identical RNA sequences would have the occasional difference due to RT during cDNA synthesis and still have its own UMI tag so artificial diversity. However, it seems that the authors are looking at homogeneous populations that defy this presumption. One way to address this question is to look at the same population twice. If the low level diversity is the result of RT error then it should not be reproducible. If it is real then it will repeat (given sufficient sampling). This is an important question both for the technique and for understanding true low level diversity in these viral populations.

PLOS authors have the option to publish the peer review history of their article (what does this mean?). If published, this will include your full peer review and any attached files.

Reviewer #1: No

Reviewer #2: No
---

## [Editor Report · Decision Letter 1]

28 Feb 2021

Dear Dr. Boritz,

We are pleased to inform you that your manuscript 'High-Throughput, Single-Copy Sequencing Reveals SARS-CoV-2 Spike Variants Coincident with Mounting Humoral Immunity during Acute COVID-19' has been provisionally accepted for publication in PLOS Pathogens.

Best regards,

Adam S. Lauring

Section Editor

PLOS Pathogens

Adam Lauring

Section Editor

PLOS Pathogens

Kasturi Haldar

Editor-in-Chief

PLOS Pathogens

orcid.org/0000-0001-5065-158X

Michael Malim

Editor-in-Chief

PLOS Pathogens

orcid.org/0000-0002-7699-2064

Nice work! Thank you for submitting to PLOS Pathogens. Your attention to the reviewers' comments is much appreciated.
---

## [Editor Report · Acceptance letter]

11 Mar 2021

Dear Dr. Boritz,

We are delighted to inform you that your manuscript, "High-Throughput, Single-Copy Sequencing Reveals SARS-CoV-2 Spike Variants Coincident with Mounting Humoral Immunity during Acute COVID-19," has been formally accepted for publication in PLOS Pathogens.

Best regards,

Kasturi Haldar

Editor-in-Chief

PLOS Pathogens

orcid.org/0000-0001-5065-158X

Michael Malim

Editor-in-Chief

PLOS Pathogens

orcid.org/0000-0002-7699-2064